# Wax Separated Effectively from Fischer-Tropsch Wax Residue by Solvent Desorption: Thermodynamic and Kinetic Analysis

**Ling Li, Yuqi Zheng, Baokang Xu, Yanhua Xu \* and Zhiying Liu \***

School of Environmental Science and Engineering, Nanjing Tech University, Nanjing 211816, China; njmuzili@njtech.edu.cn (L.L.); zhengyuqi@njtech.edu.cn (Y.Z.); baokangxu@njtech.edu.cn (B.X.)
\* Correspondence: yanhuaxu18@hotmail.com (Y.X.); zhing555@njtech.edu.cn (Z.L.)

**Abstract:** The separation and recycling of effective resources in Fischer-Tropsch wax residue (FTWR) are urgent because of the environmental hazards and energy waste they bring. In this study, organic solvents are used to separate recyclable resources from FTWR efficiently, achieving the goals of "Energy Recycle" and "Fisher-Tropsch Wax Residue Treatment". The response surface methodology (RSM) response surface analysis model accurately evaluates the relationship among temperature, residence time, liquid–solid ratio, and desorption rate and obtains the best process parameters. The results show that the product yield can reach 82.28% under the conditions of 80 °C, 4 h, and the liquid–solid ratio of 24.4 mL/g. Through the kinetic analysis of the desorption process of FTWR, the results show that the desorption process conforms to the pseudo second-order kinetic model and the internal diffusion model. The thermodynamic function results showed that there were not only van der Waals forces in the desorption process, but other strong interaction forces such as hydrogen bonds. In addition, Langmuir, Freundlich, and BET equations are used to describe the desorption equilibrium. Scanning electron microscopy (SEM) were used to analyze the pore structure of FTWR during desorption. X-ray diffraction (XRD), Fourier transform infrared spectroscopy (FT-IR), and Gas chromatography-mass spectrometer (GC-MS) analysis confirmed that the desorption product's main component was hydrocarbons (50.38 wt%). Furthermore, naphthenic (22.95 wt%), primary alcohol (11.62 wt%), esters (8.7 wt%), and aromatic hydrocarbons (6.35 wt%) compounds were found and can be further purified and applied to other industrial fields. This study shows that using petroleum ether to separate and recover clean resources from Fischer-Tropsch wax residue is feasible and efficient and has potential industrial application prospects.

**Keywords:** desorption; Fischer-Tropsch wax residue; thermodynamic; kinetic analysis

## 1. Introduction

The increasing demand for energy and the limited resources of fossil fuels have forced humans to look for renewable energy alternatives to fossil fuels [1,2]. Fischer-Tropsch synthesis (FTS) uses transition metals, such as cobalt (Co), iron (Fe), and nickel (Ni), as catalysts [3–6] to make non-petroleum raw materials (mostly coal, biomass, and natural gas). Syngas (a mixture of CO and $H_2$) is converted into pollution-free fuel and valuable chemicals. It offers a way to convert coal or natural gas into gasoline, diesel fuel, and other useful hydrocarbons such as waxes [7–10]. However, FTS technology has led to the formation of a large amount of Fischer-Tropsch wax residue (FTWR) and has caused serious industrial pollution and public land occupation problems.

FTWR is a typical wax-based solid waste mainly from the wax filter unit of FTS. Different wax filter units have various FTWR components, which are mostly composed of 40–80 wt% Fischer-Tropsch wax (FTW), 30–85 wt% Fischer-Tropsch catalyst (FTC), and 0–80 wt% minerals and other impurities [7]. In many countries, FTWR is classified as hazardous waste. If improperly handled, it will produce toxic and hazardous substances [11,12], causing damage to the ecological environment and posing a threat to

human health. Reports have reported that approximately 20,000 tons of slag wax are produced in China each year, 60% of the slag wax is incinerated, and the rest is used for landfill [13]. However, incineration and landfill treatment waste many FTW resources in FTWR, and the organic matter produced by incineration pollutes the atmosphere [14,15]. Landfill disposal will take up a lot of land resources, cause leachate to seep out, thereby polluting soil and groundwater, and will cause high economic costs and other problems [16]. Recycling the valuable resources in FTWR, avoiding resource waste, and polluting the environment is the correct way to dispose of FTWR.

Solvent desorption is a commonly used method to recover lipids from oily waste [17–19]. It has been widely used to recover aliphatic hydrocarbons from oily sludge and other oily wastes [20,21]. Mohit et al. studied the combination of MEK (polar solvent) and xylene (non-polar solvent) as a mixed solvent for the recovery of hydrocarbons from oil sludge, showing that xylene has a good recovery of hydrocarbons in petroleum efficiency [22]. Nour et al. used isopropanol solvent to extract waste engine oil to obtain regenerated oil [23]. Many studies have investigated solvent desorption to recover oily wastes containing sludge, but few have reported the recycling of FTW resources in FTWR. In fact, the current resource recovery method for wax-based solid waste is mainly pyrolysis, which converts wax into pyrolysis gas and then condenses and recovers it. The presence of the Fischer-Tropsch catalyst in FTWR affect the quality of the finished wax recovered by pyrolysis. As a result, the carbon number of the finished wax and the recovery rate were poor. This will inevitably waste a part of wax resources. The energy requirements of the pyrolysis method were also far greater than that of the solvent method.

In this study, wax-based solid waste was compared with oily solid waste, petroleum ether solvent was used to separate recycle Fischer-Tropsch wax (RFTW) from FTWR efficiently to study the influence of several factors, such as solvent/FTWR mass ratio, reaction residence time, and reaction temperature on the product yield. The product recovery rate is obtained by measuring the mass change of FTWR before and after the reaction. Central composite design-response surface methodology (CCD-RSM) was used to screen the best process conditions for FTW desorption. The desorption process of the wax in the three-way environment of Fischer-Tropsch wax, solvent and catalyst was verified and analyzed through isotherm analysis and kinetic fitting, and the desorption mechanism was proposed. Through BET, BJH, SEM morphology analysis and other characterization were used to verify the mechanism. Furthermore, the recovered products were characterized and analyzed by Fourier transform infrared spectroscopy (FT-IR), X-ray diffraction (XRD), and Gas chromatography-mass spectrometer (GC-MS). This study provides a new direction for the harmless disposal and resource utilization of FTWR. All in all, this work was devoted to finding the best extraction process and elucidating the reaction mechanism through systematic experimental and theoretical methods.

## 2. Material and Methods

FTWR was obtained from a Chinese coal company (Ningxia, China). The desorption solvent petroleum ether was provided by Sinopharm Chemical Reagent Co., Ltd. (Shanghai, China). The elemental analysis of FTWR was performed by X-ray fluorescence spectrometry (XRF, Eagle III, EDAX Inc., San Diego, CA, USA). DESIGN EXPERT 12 (Minneapolis, USA) was used to design and extract the experiment and establish a mathematical model of the relationship between each factor and the response value to determine the best research conditions.

### 2.1. Properties of FTWR

The element composition of FTWR were analyzed, and the results are shown in Table 1.

**Table 1.** The main element content of the original FTWR.

| Component | C | Fe | Si | Mn | Al | Other |
|---|---|---|---|---|---|---|
| Content (mass%) | 40.82 | 35.53 | 15.79 | 3.02 | 1.45 | 3.39 |

Note: calculated on dry basis.

### 2.2. Design of Experimental Condition Using RSM

RSM is considered a mathematical model that can accurately reveal study the relationship between various factors and response values [24–26]. It can quickly and effectively determine the best conditions for a multi-factor system [27–29]. A five-level, three-factor CCD technique was employed to optimize the independent process variables. Temperature (A), retention time (B), and liquid–solid ratio (C) in the range of 25–80 °C, 0.5–4 h, and 10–30, respectively, were considered as independent process parameters, and extracted wax yield was regarded as the desired response that needs to be maximized. Table 2 shows the coding level of the experimental variables used in the CCD method.

**Table 2.** Coding levels of experimental variables used in the CCD method.

| Name | Units | Low | High | −Alpha | +Alpha |
|------|-------|-----|------|--------|--------|
| Temperature | °C | 25 (−1) | 80 (1) | 6.2507 | 98.7493 |
| Retention time | h | 0.5 (−1) | 4 (1) | −0.693137 | 5.19314 |
| Liquid–solid ratio | mL/g | 10 (−1) | 30 (1) | 3.18207 | 36.8179 |

First, establishing a mathematical model of independent variables and response values and using Design Expert 12 software are necessary to fit the RSM model according to Equation (1) using the least square method:

$$Y = \beta_0 + \sum_{i=1}^{k}\beta_i X_i + \sum_{i=1}^{k}\beta_{ii}X_i^2 + \sum_{i>j}^{k}\sum_{j}^{k}\beta_{ij}X_iX_j, \tag{1}$$

where $Y$ signifies the predicted value of the response and $\beta_0$, $\beta_i$, $\beta_{ii}$, and $\beta_{ij}$ are the constant term and coefficient for linear, quadratic, and interaction terms, respectively, in the developed model equation. $k$ signifies the number of independent process variables chosen for process optimization ($k = 4$).

Second, ANOVA was used to analyze the optimization results, and the product desorption rate obtained in the actual experiment was used as the real value. The predicted value was obtained from the software (Design Expert 12, Minneapolis, MIN, USA). Analysis of the data through ANOVA shows the analysis of variance, the significance of variance test, and the first-order coefficient significance test regression equation. The reliability of the model is tested [28–31].

### 2.3. Solvent Extraction Experiments

To reduce the experimental error and improve the accuracy of the experiment, pre-processing the FTWR is necessary. The pretreatment steps are as follows: (1) FTWR was putted in a drying oven and dry it at 80 °C for 8 h; (2) the dried FTWR was grinded and crushed into small particles and pass it through a 120-mesh screen to obtain a FTWR with a particle size of less than 134 μm. FTWR after grinding and sieving was used in the following experiments. Typically, the desorption experiment was carried out according to the RSM response surface optimization design plan. The required mass of FTWR is accurately weighed, FTWR wrapped with filter paper, and placed in a 100 mL flask. The necessary volume of petroleum ether is then added to the flask. The flask was later placed on a constant temperature shaker (SHA-B, GUOHUA, Changzhou, China) to complete the design conditions' entire reaction process. The solvent in the washed extract was evaporated on a rotary evaporator. Finally, the residue was weighed and named as Recycle Fischer-Tropsch wax (RFTW). The residues after FTWR leaching were dried and weighed to calculate the product yield. Equation (2) was used to calculate the yield of the extracted product in FTWR. Each experiment was repeated three times, and the results were averaged.

$$Y_{exp} = \frac{M - m}{\alpha M} \times 100\% \tag{2}$$

where $Y_{exp}$ represents the recovery rate of RFTW, $M$ and $m$, respectively, represent the mass of FTWR before and after treatments (g), $\alpha$ represents the mass percentage of carbon-containing compounds in FTWR obtained by XRF analysis, and $\alpha = 0.4082$ in this experiment.

*2.4. Material Characterization*

Specific surface area and detailed porosity parameters (pore size distribution and pore volume) were determined using the Brunauer–Emmett–Teller (BET) and the Barrett–Joyner–Halenda plot methods. The nitrogen adsorption and desorption curves of FTWR were also obtained. Morphology was characterized by using a scanning electron microscope (SEM; S4800, Hitachi Corporation, Tokyo, Japan). An X-ray diffractometer (Thermo Fisher Scientific Co., Ltd., Waltham, MA, USA) was used to collect XRD data of RFTW with monochromatic Cu/K$\alpha$ radiation ($\lambda$ = 1.54056 Å) at 36 kV and 20 mA in a 2 θ scanning range of 20–70°. The types of functional groups in RFTW were analyzed by NEXUS870 Fourier Transform Infrared Spectrometer (FT-IR) (Nicolet 8700, Thermo Fisher Scientific, Waltham, USA). The scanning was done from 400 cm$^{-1}$ to 4000 cm$^{-1}$ at a scanning rate of 40 with the step size of 4 cm$^{-1}$ by adopting attenuated total reflectance. Gas chromatography-mass spectrometry (GC-MS) (Shimadzu Corporation, Tokyo, Japan) was used to analysis the components of samples. The capillary column used VF-5MS (30 m $\times$ 0.25 mm; 0.25 μm), the carrier gas was helium, and the flow rate was 0.8 mL/min. The analysis was performed in duplicate by injecting 1 μL in splitless mode. The sampler was set to 280 °C, the transfer line was set to 280 °C, and the ion source was set to 300 °C. The column was initially set to 100 °C for 1 min, heated at 10 °C/min to 200 °C and 200 °C for 2 min, and finally reheated at 3.5 °C/min until 260 °C. The total analysis time was 40 min. The mass spectrum was collected every 0.5 s with a range of $32-380$ $m/z$. In this way, qualitative and quantitative analysis of RFTW was carried out.

## 3. Results and Discussions

### 3.1. RFTW Desorption Efficiency from FTWR

3.1.1. Statistical Analysis

The matrix for different experimental conditions was presented in Table 3. According to the design matrix, a total of 20 sets of experiments were designed. Table 3 showed the experimental and predicted values corresponding to different experimental conditions.

**Table 3.** Experimental responses and predicted responses through response surface methodology.

| Run Order | Temperature (°C) | Retention Time (h) | Liquid–Solid Ratio (mL/g) | RFTW Yield (wt%) | |
|---|---|---|---|---|---|
| | | | | **Experimental Value** | **Predicted Value** |
| 1 | 50 | 0.5 | 30 | 20.66 | 26.18 |
| 2 | 50 | 2 | 20 | 54.97 | 54.16 |
| 3 | 50 | 2 | 20 | 49.71 | 54.16 |
| 4 | 80 | 0.5 | 20 | 55.59 | 55.25 |
| 5 | 25 | 0.5 | 20 | 28.22 | 20.45 |
| 6 | 80 | 3 | 10 | 74.68 | 72.05 |
| 7 | 50 | 2 | 20 | 49.69 | 54.16 |
| 8 | 25 | 2 | 30 | 33.05 | 32.28 |
| 9 | 25 | 2 | 10 | 34.85 | 33.63 |
| 10 | 50 | 4 | 10 | 55.11 | 52.18 |
| 11 | 50 | 4 | 30 | 61.20 | 61.59 |
| 12 | 50 | 4 | 15 | 57.17 | 58.57 |
| 13 | 25 | 4 | 20 | 42.54 | 46.45 |
| 14 | 50 | 2 | 15 | 55.80 | 53.15 |
| 15 | 80 | 3 | 30 | 76.20 | 76.08 |
| 16 | 80 | 4 | 20 | 83.05 | 81.26 |
| 17 | 50 | 2 | 20 | 55.43 | 54.16 |
| 18 | 50 | 2 | 20 | 54.86 | 54.16 |
| 19 | 50 | 2 | 20 | 53.38 | 54.16 |
| 20 | 50 | 0.5 | 10 | 33.51 | 35.58 |

Externally studentized residuals were used to check regression assumptions, which will map all the different normal distributions to a single standard normal distribution and make it more sensitive to find problems with the analysis. Figure 1a was a normal probability plot, which indicated whether the residuals followed a normal distribution. Figure 1b was a plot of the residuals versus the ascending predicted response values. Figure 1c was a plot of the residuals versus the experimental run order. It checked for lurking variables that may have influenced the response during the experiment. Figure 1d was a plot of residuals versus temperature. It checked whether the variance not accounted for by the model is different for different levels of temperature. The graph exhibited a random scatter, which indicates that everything is normal. Figure 1e showed the predicted response values versus the experimental response values. The 45° line represents the expected value, and the dot represents the experimental value obtained by the actual experiment [32]. The figure showed that the two are close, which means that the obtained model can better represent the real experimental data [33,34].

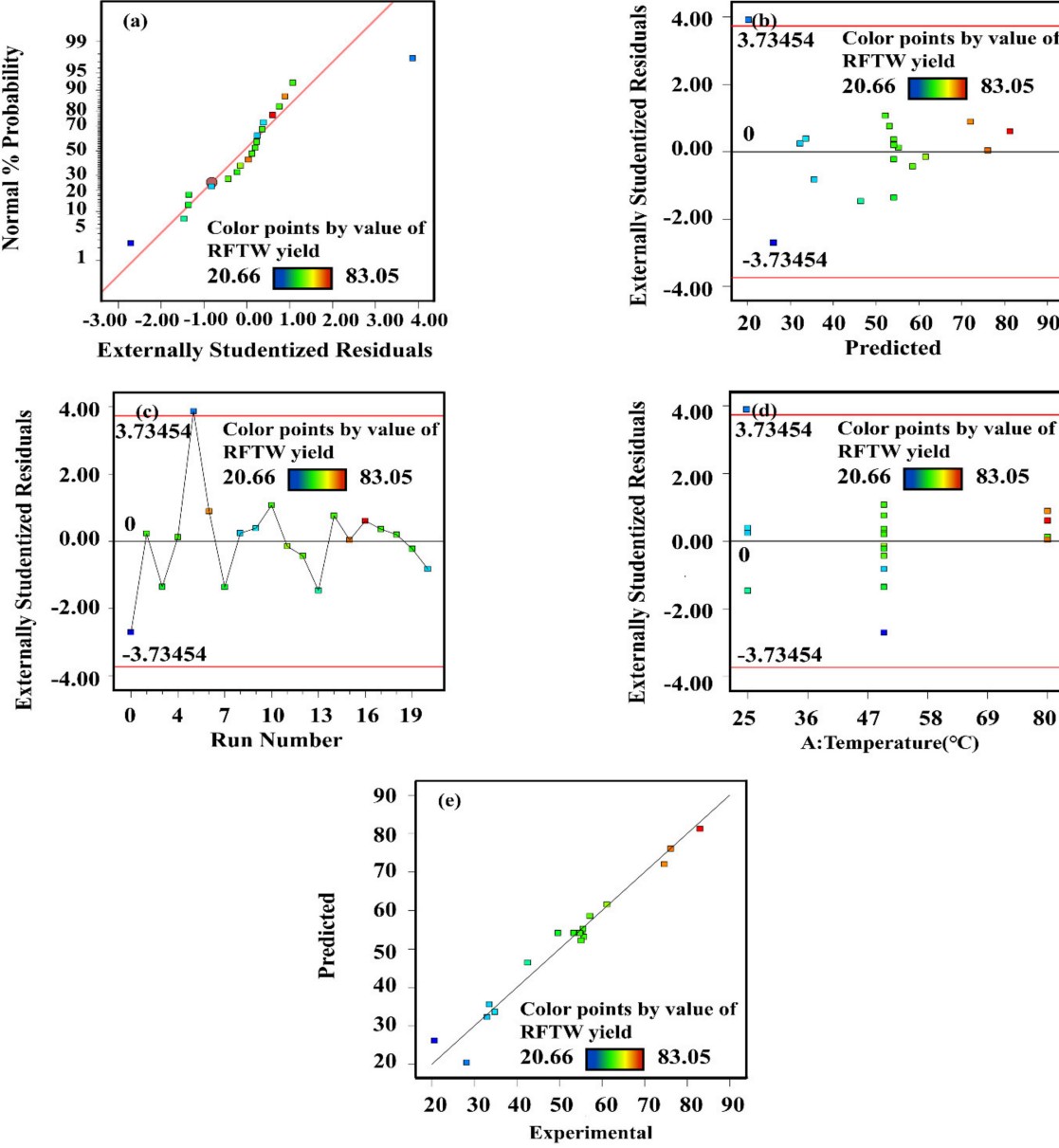

**Figure 1.** (**a**) Normal plot of residuals, (**b**) residuals vs. predicted, (**c**) residuals vs. run, (**d**) residuals vs. A: temperature (°C), (**e**) predicted vs. experimental.

Table 4 showed the analysis of variance of the desorption rate regression model. The model F-value of 71.59 implied that the model is significant. The chance that an F-value this large could occur due to noise is 0.01%. *p*-values less than 0.0500 indicated that model terms are significant. In this case, A, B, BC, $B^2$, and $C^2$ are significant model terms. Values greater than 0.1000 indicate that the model terms are insignificant. The lack of fit F-value of 2.35 implies the lack of fit is not significant relative to the pure error. The chance that a lack of fit F-value this large could occur due to noise is 18.03%. Non-significant lack of fit is good. Table 5 shows the correlation coefficient of the statistical model. The predicted $R^2$ of 0.8992 was in reasonable agreement with the adjusted $R^2$ of 0.9489 (i.e., the difference is less than 0.2). Adeq precision measures the signal to noise ratio. A ratio greater than 4 was desirable. A ratio of 30.650 indicated an adequate signal. This model can be used to navigate the design space [35].

**Table 4.** ANOVA of RSM for the reduced quadratic model.

| Source | Sum of Squares | Degree of Freedom | Mean Square | F-Value | *p*-Value | Remark |
|---|---|---|---|---|---|---|
| Model | 4696.99 | 5 | 939.40 | 71.59 | <0.0001 | significant |
| A—temperature | 2385.34 | 1 | 2385.34 | 181.79 | <0.0001 | significant |
| B—retention time | 1544.16 | 1 | 1544.16 | 117.68 | <0.0001 | significant |
| BC | 102.01 | 1 | 102.01 | 7.77 | 0.0145 | significant |
| $B^2$ | 206.05 | 1 | 206.05 | 15.70 | 0.0014 | significant |
| $C^2$ | 127.88 | 1 | 127.88 | 9.75 | 0.0075 | significant |
| Residual | 183.70 | 14 | 13.12 | - | - | - |
| Lack of fit | 148.53 | 9 | 16.50 | 2.35 | 0.1803 | not significant |
| Pure error | 35.17 | 5 | 7.03 | - | - | - |
| Cor total | 4880.69 | 19 | - | - | - | - |

**Table 5.** Correlation coefficient of the statistical model.

| Std. Dev. | 3.62 | $R^2$ | 0.9624 |
|---|---|---|---|
| Mean | 51.48 | Adjusted $R^2$ | 0.9489 |
| C.V. % | 7.04 | Predicted $R^2$ | 0.8992 |
| - | - | Adeq precision | 30.6502 |

The polynomial equation between response (yield) and variables (temperature (A), retention time (B), and liquid–solid ratio (C)) is given by Equation (3).

$$Y = 57.74 + 17.40A + 13.00B + 4.70BC - 6.89B^2 - 5.39C^2 \tag{3}$$

### 3.1.2. The Effect of RSM-Based Process Parameters on Desorption Efficiency

As shown in Figure 2, the 2D plane contour map and the 3D response map formed by the fitting equation can intuitively show the influence of temperature, residence time, and liquid–solid ratio on the desorption rate. As the desorption time increases, the higher the temperature, the higher the yield of RFTW, and then gradually stabilizes, when the solid–liquid ratio is constant. This showed that the reaction is an endothermic reaction, and the temperature rise is conducive to the thermal movement of FTW molecules, thereby destroying the confinement effect of the pores in the FTC. When the temperature was kept constant, the liquid–solid ratio increases. As the desorption time increases, the desorption efficiency of RFTW first increases and then decreases. It achieved its peak when the liquid–solid ratio is 24.4:1 mL/g. This attributed to the particle concentration effect that limits the continued improvement in efficiency [36]. Figure 2b showed the coupling effect of the fixed temperature at 80 °C, residence time, and liquid–solid ratio on the product yield. The optimization analysis was carried out through Design Expert 12 software.

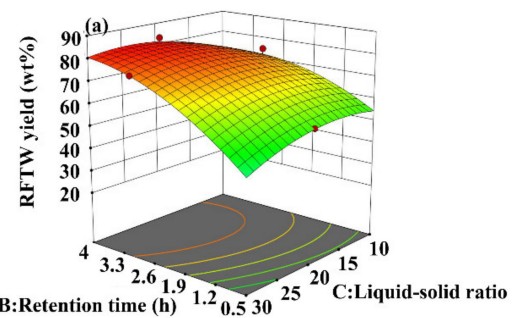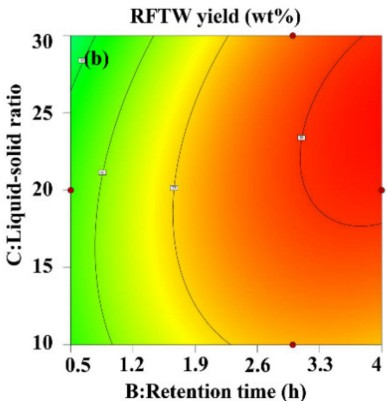

**Figure 2.** (**a**) Response surface of the influence of B and C on the RFTW yield, (**b**) contours of the influence of B and C on the RFTW yield.

Table 6 gives independent variables and response as constraints for optimization. Given the three indicators, the process parameters with the highest desorption rate are predicted as follows: 80 °C, 4 h, and liquid–solid ratio 24.4 mL/g. Under these conditions, the product yield could reach 82.28%.

**Table 6.** Independent variables and response as constraints for optimization.

| Parameters | Objective | Lower Limit | Upper Limit |
|---|---|---|---|
| Temperature (°C) | In range | 25 | 80 |
| Retention time (h) | In range | 0.5 | 4 |
| Liquid–solid ratio (mL/g) | In range | 10 | 30 |
| Extracted wax yield (wt%) | Maximum | 20.66 | 100 |

### 3.1.3. Verification of the Best Process Parameters

Given experimental operability, the optimal process parameters were optimized as follows: 80 °C, 4 h, and the liquid–solid ratio of 24. The experiment was verified under the best process conditions. The results were shown in Table 7. The average error range from the predicted value is 1.68%, which is acceptable.

**Table 7.** Experimental and predicted values of RFTW yield at optimum condition.

| Run | Temperature (°C) | Retention Time (h) | Liquid–Solid Ratio (mL/g) | RFTW Yield (wt %) | | Error (%) |
|---|---|---|---|---|---|---|
| | | | | Experimental | Predicted | |
| 1 | 80 | 4 | 24 | 83.54 | 82.28 | 1.51 |
| 2 | 80 | 4 | 24 | 84.12 | 82.28 | 2.19 |
| 3 | 80 | 4 | 24 | 83.41 | 82.28 | 1.35 |
| Average | - | - | - | 83.69 | 82.28 | 1.68 |

### 3.2. Desorption Isotherms Analysis

In general, adsorption behavior occurs simultaneously along with desorption behavior and their reaction rates are equivalent at equilibrium. Therefore, some adsorption models can be applied to the desorption system of FTWR. In this study, the Langmuir and Freundlich models were used to fit the equilibrium data in the FTWR desorption process. The relevant equations are as follows [37].

$$q_e = q_m k_L \frac{C_e}{1 + k_L C_e} \tag{4}$$

$$q_e = k_{LF} C_e^{1/n} \tag{5}$$

where $q_m$ and $k_L$ are Langmuir constants of maximal uptake (mg/g$^{-1}$) and desorption coefficient (g/mL), $k_{LF}$ and $n$ are two Freundlich isotherm constants.

The fitted isotherms of Langmuir and Freundlich models are presented in Figure 3, and the isotherm correlated parameters are shown in Table 8. The correlation coefficients $R^2$ fitted by Langmuir and Freundlich equations were all less than 0.99, which means that the fitting effect was not ideal. In addition, the amount of desorption when the Langmuir model reached equilibrium is less than the experimental value, while B is the opposite. Both Langmuir and Freundlich isotherm models assume that the adsorbent surface is a uniform monolayer adsorption, which is contrary to the pore structure and non-uniform surface of FTC. Therefore, the Langmuir and Freundlich equations are not suitable for simulating the desorption process of FTWR.

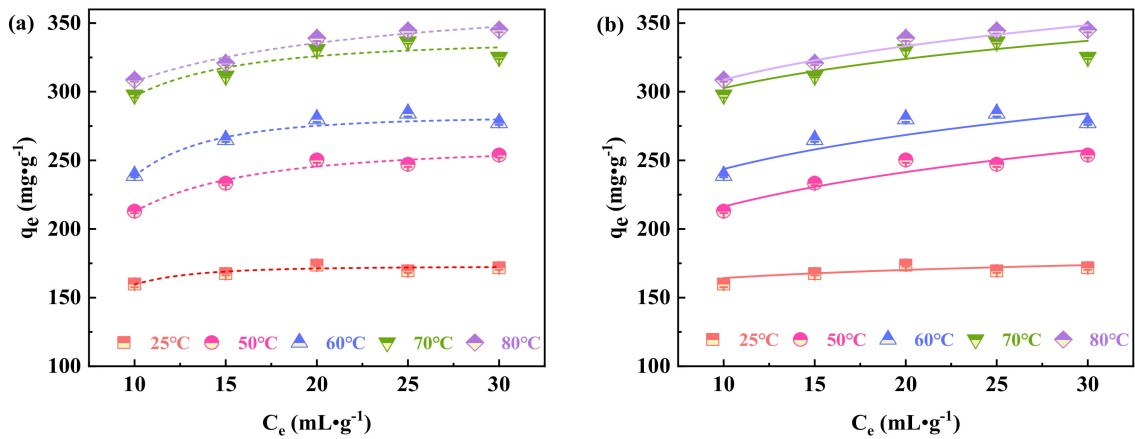

**Figure 3.** Desorption isotherms of FTWR at different temperatures, (**a**) Langmuir, (**b**) Freundlich.

**Table 8.** Adsorption isotherm model constants and correlation coefficients of FTWR.

| Temperatures | Langmuir | | | Freundlich | | |
|---|---|---|---|---|---|---|
| | $R^2$ | $K_L$ (g/mL) | $q_m$ (mg/g) | $R^2$ | $K_F$ | $n$ |
| 25 °C | 0.744 | 0.005 | 172.6 | 0.882 | 146.090 | 9.638 |
| 50 °C | 0.928 | 0.123 | 260.4 | 0.895 | 150.152 | 6.307 |
| 60 °C | 0.949 | 0.007 | 282.0 | 0.846 | 176.494 | 7.139 |
| 70 °C | 0.843 | 0.089 | 338.0 | 0.674 | 241.505 | 10.202 |
| 80 °C | 0.922 | 0.658 | 383.5 | 0.954 | 239.577 | 9.079 |

In order to study the desorption process of FTWR better, this research continued to study the nature of the pores in FTWR based on the $N_2$ adsorption-desorption analysis. It could be seen from Figure 4 that the average pore diameter of Fresh FTWR was almost 0 and the pore volume is negative, which means that almost no pores were exposed in Fresh FTWR, that is, FTW almost blocked all the pore structure of FTC. As the solvent residence time increases, the average pore diameter changes from larger to smaller, and the cumulative pore volume gradually increases. The average pore size gradually increased to 32 nm in the first 1 h, and then slightly decreased for 1–4 h, the average pore size was 29 nm at 4 h. This means that during the desorption process, pores with larger pore diameters were desorbed first; then more small pores were exposed, resulting in a gradual decrease in the average pore diameter.

Figure 5 showed the nitrogen adsorption and desorption curve. The shape of the adsorption equilibrium isotherm was related to the pore structure of the material. The amount of desorbed gas decreased with the decrease of component partial pressure. The downward trend of the curve indicated that the desorption rate was faster in the early stage, which is due to the lower desorption heat of the multi-molecular layer. FTW is more

difficult to desorb in the later stage of desorption, as the desorption process progresses, which attributed that the desorption heat of the bottom layer is less than the heat of liquefaction of FTW. The $N_2$ adsorption–desorption isotherms of FTWR at different times belong to type III adsorption isotherms [38]. This indicated that the interaction between the solid and the adsorbate was smaller than the interaction between the adsorbates. The pore size distribution diagram at Figure 5 (inner) which detailed showed the pore size distribution of the fresh FTWR and the FTWR during the extraction process, which is consistent with the pore size change described in Figure 4.

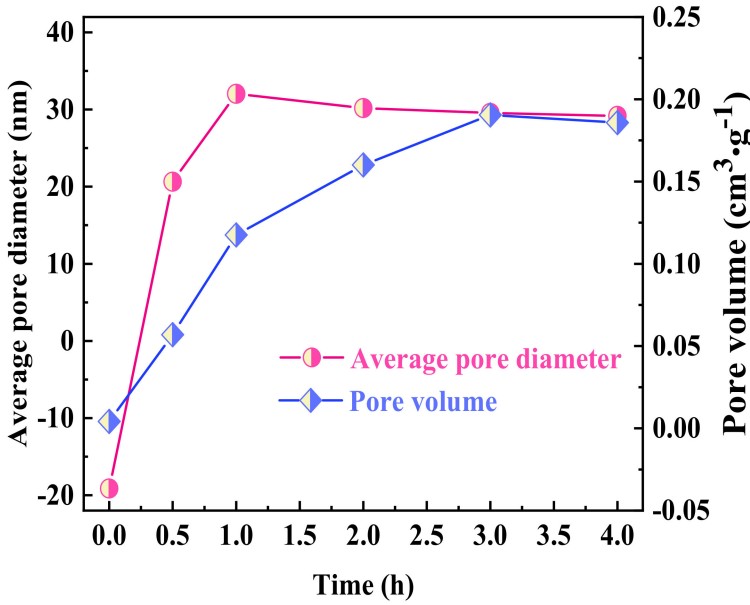

**Figure 4.** Variation of average pore width (**left**) and single point adsorption total pore volume (**right**) with time in desorption process of FTWR.

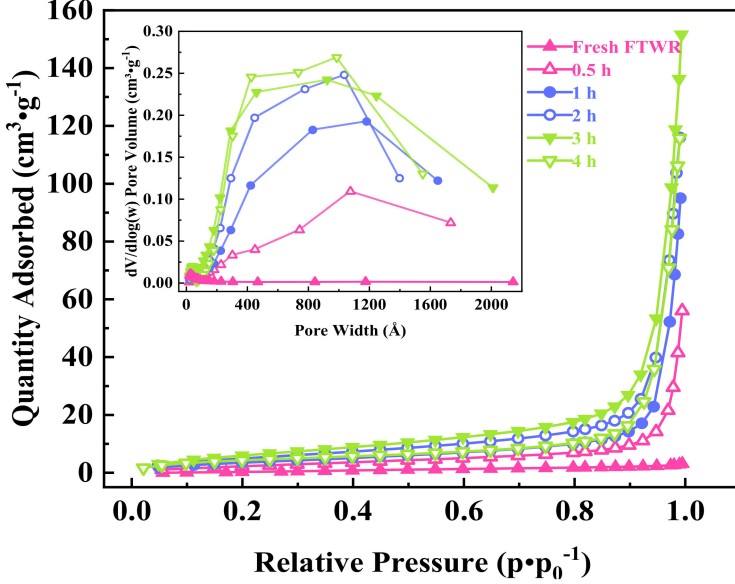

**Figure 5.** $N_2$ adsorption–desorption isotherms and pore size distribution (**inner**) of FTWR.

### 3.3. FTWR Desorption Thermodynamic Analysis

A thermodynamic analysis was performed to further reveal the nature of the desorption behavior. The thermodynamic parameters of the desorption process include Gibbs free energy change ($\Delta G$), enthalpy change ($\Delta H$), entropy change ($\Delta S$), and the related calculation equations are as follows [37]:

$$K_C = \frac{C_e}{q_e} \tag{6}$$

$$\Delta G = -RT \ln K_C \tag{7}$$

$$\ln K_C = \frac{\Delta S}{R} - \frac{\Delta H}{RT} \tag{8}$$

where $K_C$ is the thermodynamic equilibrium constant, and R and T represent the universal gas constant (8.314 J/mol K) and solution temperature (K), respectively. The values of $C_e$ and $q_e$ are obtained from Equation (4), the value of $\Delta G$ is obtained from Equation (7), and the values of $\Delta H$ and $\Delta S$ are determined by the slope and intercept of the curve fitting in Figure 6.

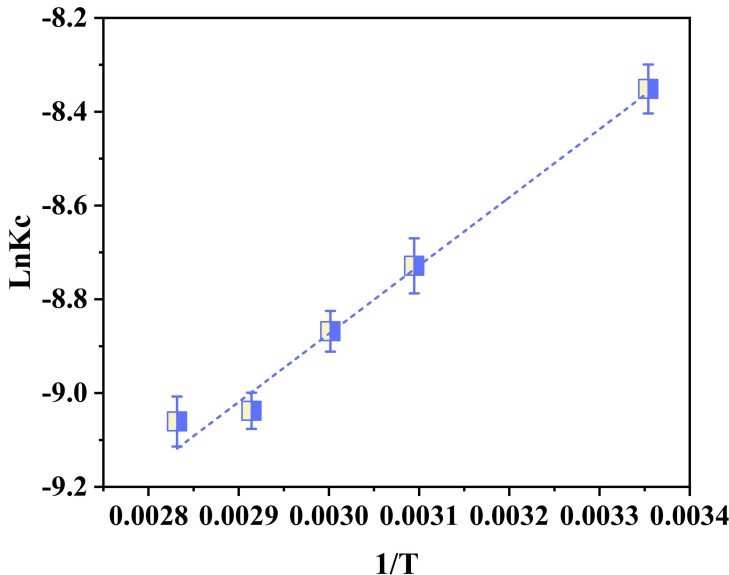

**Figure 6.** The plot of lnKc vs. 1/T for thermodynamic parameters determination.

The thermodynamic parameter values of these desorption processes are summarized in Table 9.

**Table 9.** Thermodynamic parameter value.

| T (°C) | $\Delta G$ (kJ·mol$^{-1}$) | $\Delta H$ (kJ/mol) | $\Delta S$ (J/mol K) |
|---|---|---|---|
| 25 | −20.702 | | |
| 50 | −23.451 | | |
| 60 | −24.563 | 12.092 | 110.049 |
| 70 | −25.784 | | |
| 80 | −26.603 | | |

The experimental data of $\Delta G < 0$ showed that the desorption of organic matter from the particles was a spontaneous process, and its size mainly depends on the strength of adhesion. The value of $\Delta H$ was 12.092 kJ/mol, which showed that the reaction was an endothermic process, which also explains why the increase in temperature is beneficial to the desorption of organic matter. A positive value of $\Delta S$ indicates that the confusion of

the whole reaction has increased. During the desorption process, the vacancies left by the organic molecules will be occupied by solvent molecules. In addition, the positive value of ΔS also showed that was not only van der Waals force in the desorption system, but other strong forces such as hydrogen bonds or chemical bonds could exist.

### 3.4. FTWR Desorption Kinetic Analysis

The pseudo-first-order, pseudo-second-order, and intraparticle diffusion models were used to evaluate the desorption kinetics of RFTW on FTWR and study the possible mechanism of its desorption process. The correlated equations are as follows [39].

$$\ln(q_e - q_t) = \ln q_e - k_1 t \tag{9}$$

$$\frac{t}{q_t} = \frac{1}{k_2 q_e{}^2} + \frac{t}{q_e} \tag{10}$$

$$q_t = k_i t^{1/2} + C \tag{11}$$

where $q_e$ (mg/g) and $q_t$ (mg/g) represent the quantity of RFTW desorbed per unit mass of FTWR (g) at equilibrium and varying t (h), respectively. $k_1$ (h$^{-1}$) and $k_2$ (h$^{-1}$) are the pseudo-first-order and pseudo-second-order rate constants, respectively. C is a constant related to the thickness and boundary layer, and $k_i$ is the internal diffusion rate constant. The corresponding kinetic fitting parameters and correlated coefficients were listed in Table 10. The values of $q_e$, k and C can be determined by the slope and intercept of the fitted curve.

**Table 10.** First and second-order kinetics and intraparticle diffusion model fitting dynamics equation of different temperatures.

| Models | | 25 °C | 50 °C | 60 °C | 70 °C | 80 °C |
|---|---|---|---|---|---|---|
| $q_{exp}$ (mg/g) | | 183.5 | 326.3 | 323.1 | 364.6 | 372. 4 |
| pseudo-first-order | $R^2$ | 0.832 | 0.920 | 0.895 | 0.863 | 0.923 |
| | $k_1$ (h$^{-1}$) | 1.656 | 0.811 | 1.310 | 1.584 | 1.811 |
| | $q_e$ (mg/g) | 161.9 | 238.3 | 274.2 | 316.5 | 329.2 |
| pseudo-second-order | $R^2$ | 0.999 | 0.990 | 0.997 | 0.994 | 0.999 |
| | $K_2$ (h$^{-1}$) | 0.012 | 0.003 | 0.005 | 0.006 | 0.007 |
| | $q_e$ (mg/g) | 184.5 | 324.7 | 322.6 | 363.6 | 370.4 |
| intraparticle diffusion model | $R^2$ | 0.946 | 0.945 | 0.967 | 0.978 | 0.951 |
| | $K_i$ (h$^{-1}$) | 31.618 | 44.814 | 78.088 | 73.093 | 60.354 |
| | C | 104.939 | 112.194 | 130.953 | 186.918 | 220.907 |

It could be seen from Table 9 that the correlation coefficients $R^2$ fitted by the pseudo-first-order kinetic model are all less than 0.99, and the maximum desorption amounts when the desorption equilibrium is reached at 25–80 °C are 161.939, 238.333, 274.230, 316.496, and 329.212 mg/g, respectively. Both are less than the desorption amount in actual equilibrium. This showed that the desorption process of FTWR does not conform to the pseudo-first order kinetics. The correlation coefficients of the pseudo-secondary kinetic model fitting except for 50 °C (0.990), the $R^2$ of the other temperature fitting models are all greater than 0.99, and the maximum desorption capacity when the desorption equilibrium is reached at 25–80 °C are 183.453, 326.272, 323.090, 364.616, and 372.381 mg/g, respectively, are close to the actual experimental value. The results indicated that the pseudo-second-order kinetic model can better describe the desorption process, which indicated that there are adsorption-desorption saturation sites, and the reaction has multiple complex effects. Therefore, the internal diffusion model was used to continue the fitting. In Figure 7c, $q_t$ was plotted against $t^{1/2}$, and each temperature fitting line does not pass through the origin (intercept C = 0), which indicated that the internal diffusion was not controlled by a single rate. In Figure 7d, the internal diffusion model fitting graph can be fitted into three linear parts, and the fitting correlation coefficient $R^2$ is all greater than 0.9, which showed that the desorption of FTWR was divided into three stages: the desorption rate in the first stage was increased slowly; the desorption rate in the second stage was accelerated; and the desorption rate in the third stage was tended to be flat. The first stage was related to surface diffusion,

namely waste Fischer-Tropsch catalyst (WFTC), the FTW adsorbed on the surface layer was desorbed in the solvent; the second stage was the intra-particle diffusion process, that is, the FTW deposited in the spent catalyst pores was desorbed by the solvent; the third stage was the dynamic equilibrium process of adsorption and desorption, at this time the inside of the particle diffusion becomes very weak.

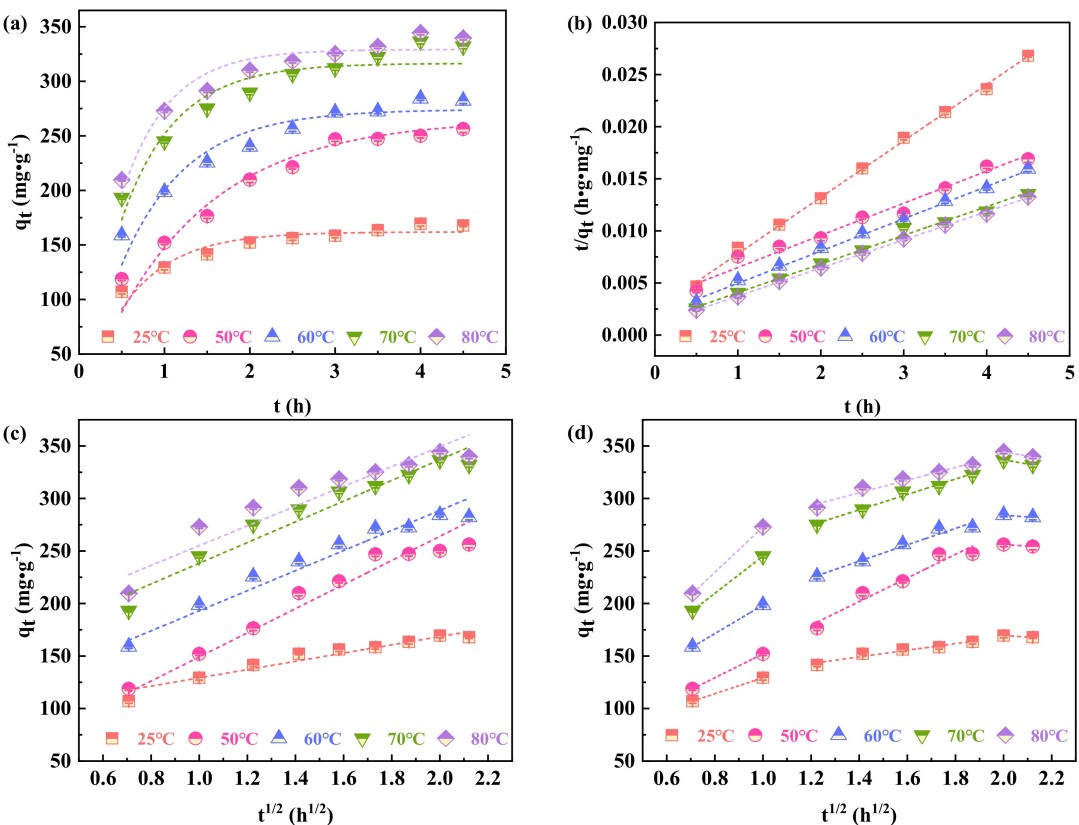

**Figure 7.** (**a**) Pseudo-first-order, (**b**) pseudo-second-order, (**c**) linear fitting of intraparticle diffusion model, (**d**) piecewise linear fitting of intraparticle diffusion model.

### 3.5. Changes in Surface Morphology during Desorption

The SEM results of WR in different time periods show that the pore structure in WR gradually appears as time increased. As shown in Figure 8a, the surface of the fresh WR is covered with a layer of wax without any pore structure. In Figure 8b,c, the pores gradually became more and more clear. The result is also consistent with the description in Figure 4.

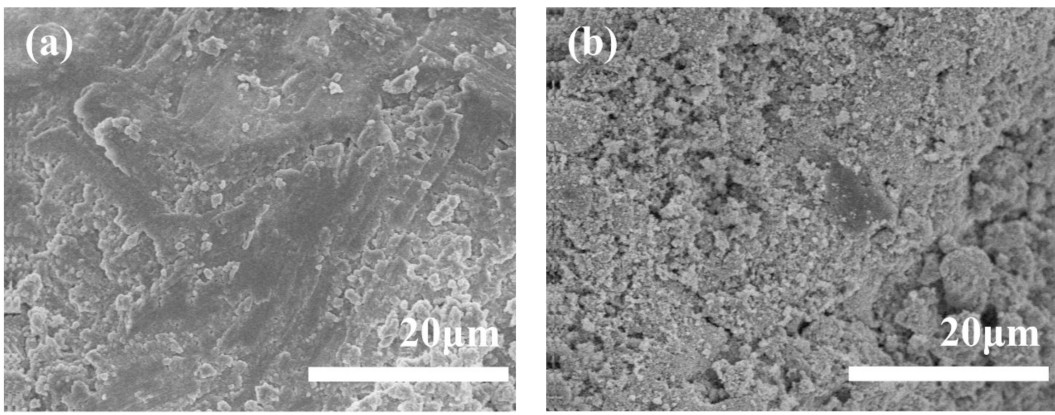

**Figure 8.** *Cont.*

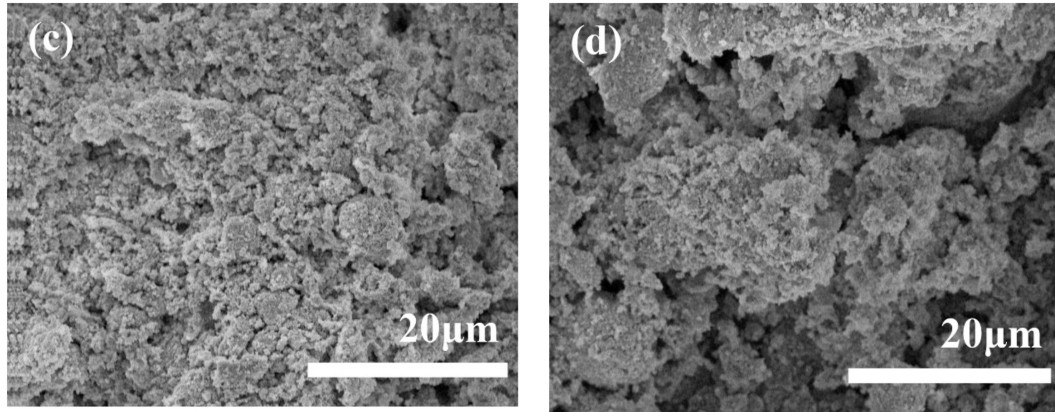

**Figure 8.** SEM images of the FTWR (**a**) fresh FTWR, (**b**) 1 h, (**c**) 2 h, (**d**) 4 h.

### 3.6. FTWR Desorption Mechanism Analysis

According to kinetic analysis and SEM morphology analysis, it was speculated that the desorption process of RFTW in FTWR is shown in Figure 9. As shown in Figure 9a,b, the porous structure and rich specific surface area of FTC make it easier for small molecules in FTC, such as saturated light hydrocarbons and aromatic compounds, to enter the pores of FTC. After the pores were blocked by small molecular compounds, more FTW could only be attached to the surface of the FTC. Attributed to the high temperature of the Fischer-Tropsch reaction system, FTW was in a fluid state, and a large amount of FTW and FTC were bonded and wrapped to form WFTR. Figure 9c,d depicted the desorption process of FTW in FTWR. Owing to the principle of similar compatibility, the FTW on the surface of the FTC was first desorbed into the solvent, which is related to the surface diffusion of the particles, and the desorption rate at this stage was relatively slow. The second stage was related to the intra-particle diffusion. The interaction between FTW molecules was greater than the effect of pores on FTW molecules, and FTW desorption in the pores of the spent catalyst. According to the desorption isotherm and kinetic analysis, this stage was multi-molecular layer desorption, with lower heat of desorption and faster desorption rate. The desorption rate in the third stage was relatively gentle, which is because the heat of desorption of the underlying molecules was less than the heat of liquefaction of FTW. After reaching the desorption equilibrium, RFTW and WFTC were obtained.

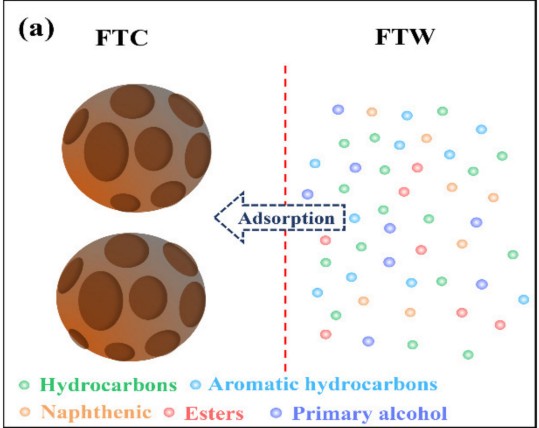
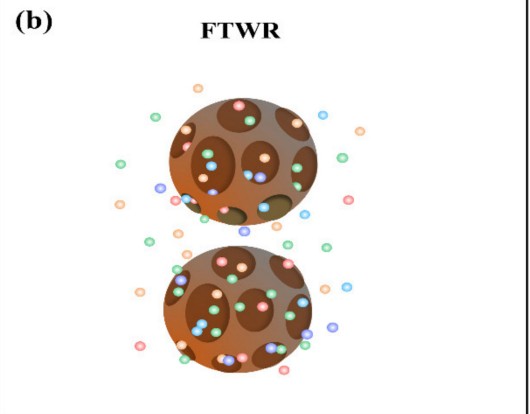

**Figure 9.** *Cont.*

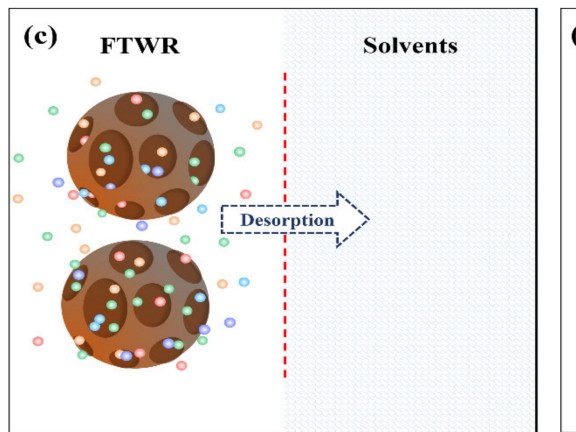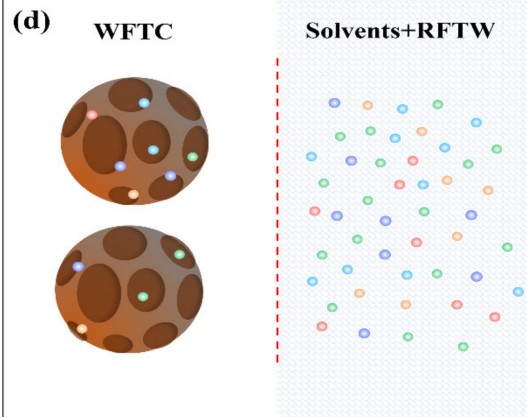

**Figure 9.** Schematic diagram of the desorption process of FTWR: adsorption process (**a**,**b**) and desorption process (**c**,**d**).

### 3.7. Characteristics of Extracted Product

3.7.1. XRD Diffraction Analysis

The XRD patterns of the product (RFTW) obtained under the optimal process conditions and fresh FTWR were showed in Figure 10. The characteristic peaks of the XRD pattern of Fresh FTWR are not obvious, and the intensity was weak, which may be on account of the low crystallinity of fresh FTWR. Obvious characteristic peaks could be seen in the XRD pattern of RFTW after extraction. The XRD pattern of the RFTW showed that the diffraction peaks at 21.6°, 23.9°, 30.0°, and 36.1° correspond to (110), (200), (210), and (020) crystal planes, respectively. The untreated FTWR has no obvious peaks. The PDF card of $(CH_2)_x$ has the highest matching degree. Apparently, RFTW contains amounts of olefins. However, taking into account the complex composition of the system, one could assume that this is a mixture of crystalline and amorphous phases.

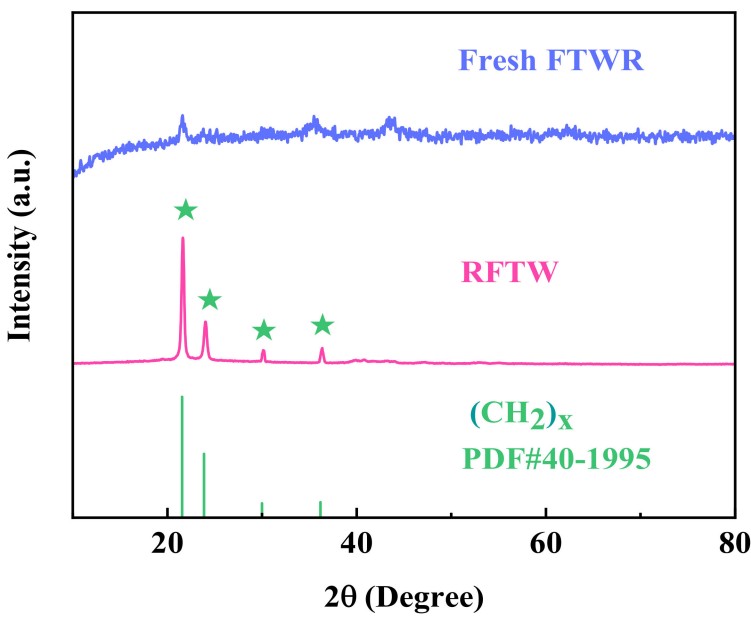

**Figure 10.** X-ray diffraction pattern fresh FTWR and RFTW.

3.7.2. IR Analysis

As shown in Figure 11, the IR spectrums illustrate that the characteristic peak at 3406 cm$^{-1}$ is attributed to the O-H deformation vibrations, respectively. The characteristic peaks bands at 2921 and 2852 cm$^{-1}$ attributed to the symmetrical and asymmetrical stretching of hydrocarbons C-H in CH$_2$ and CH$_3$ groups. Deformation vibrations contributed by

related to the vibrations of benzene derivatives can be observed at 1893 cm$^{-1}$. There is one peak at 1748 cm$^{-1}$, which can be assigned to ester carbonyl (C=O) group stretching vibrations (triglycerides). Peaks bands at 1462 and 1371 cm$^{-1}$ attributed to CH$_2$ and CH$_3$ aliphatic groups (scissoring vibrations). Notably, the peak at 721 cm$^{-1}$ is related to the CH$_2$ rocking vibration of cis-di-substituted olefins [40–42].

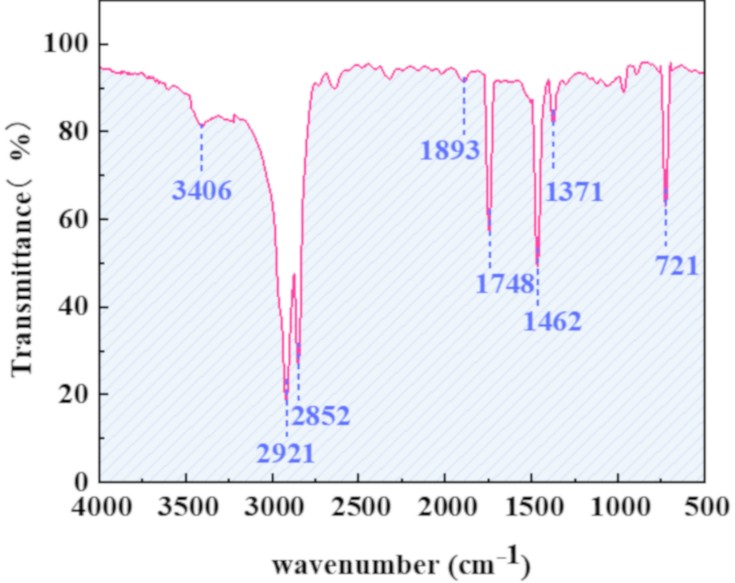

**Figure 11.** IR spectrum of RFTW.

### 3.7.3. GC-MS Analysis

GC-MS analysis was performed to detect the chemical composition of the product more clearly. Figure 12 shows the product distribution at different residence times. In the first 5 min, naphthenic was the main one, followed by aromatic hydrocarbons, and amounts of esters and primary alcohol were produced in the middle 15–30 min. Hydrocarbons mainly focuses on 20–23 and 30 min later. Therefore, it can be considered that the main components extracted product were saturated hydrocarbons, naphthenes, esters, primary alcohols, and aromatic compounds.

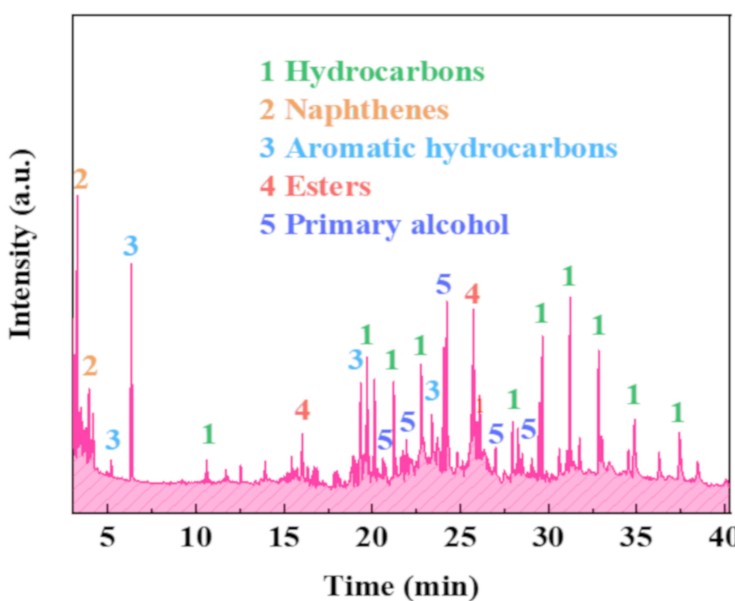

**Figure 12.** GC-MS analysis total ion spectrum.

Figure 13 showed each component's relative content (select components with content greater than 1%) and the total content of each category. Among them, the highest content was saturated hydrocarbons compounds, accounting for 50.38 wt%. Among all the straight-chain alkanes, $C_{19}H_{40}$ has the highest content (24.51 wt%), which is the raw material for producing high-quality diesel. Furthermore, the product also contains 22.95 wt% naphthenes, 11.62 wt% primary alcohols, 8.7 wt% polar esters, and 6.35 wt% aromatic compounds. Primary alcohol compounds and polar wax esters are important raw materials for various industries, including candles, wood board sizing agents, lubricants, coatings, packaging, food, and cosmetics industries [43,44]. Apparently, the RFTW has a good industrial application prospect.

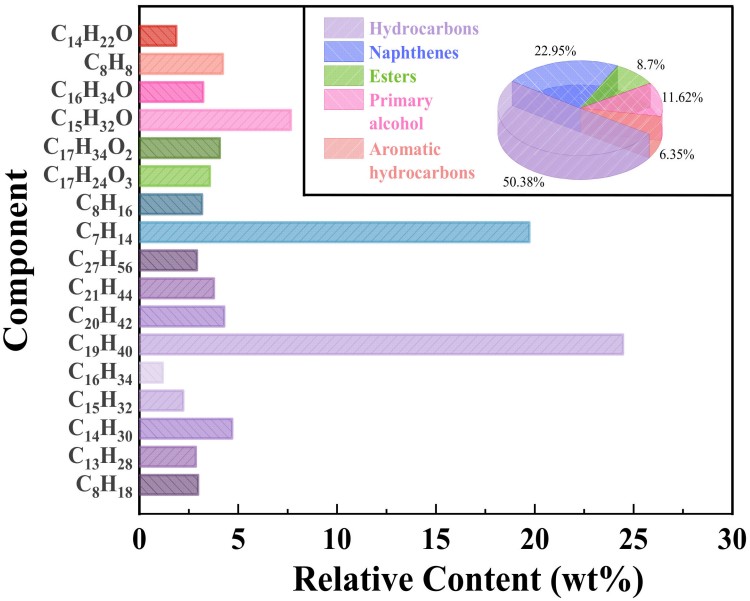

**Figure 13.** The relative content of each component (>1%) and the total content of each category.

## 4. Conclusions

The wax from FTWR was obtained by solvent extraction with petroleum ether. Through the RSM mathematical model prediction, the optimal process conditions were obtained, and the actual experiment verified that the product yield under the optimal reaction conditions could reach 82.28%. As the temperature increased, it was beneficial to the thermal movement of FTW molecules, thereby destroying the limiting effect of FTC and increasing the recovery rate. The recovery rate decreased with the increase of the liquid–solid ratio, which was attributed to the particle concentration effect which limited the continuous increase of the recovery rate. In addition, the petroleum ether solvent has similar compatibility with the recovered wax, which was conducive to the desorption process.

The $N_2$ adsorption-desorption isotherm could better define the desorption process, indicated the multi-molecular layer adsorption and desorption behavior of FTWR. The thermodynamic function results showed that there were not only van der Waals forces in the desorption process, but other strong interaction forces such as hydrogen bonds. In addition, the desorption process could be well described by quasi-second-order kinetics and internal diffusion models. This indicated that the desorption behavior on FTWR was mainly controlled by chemical desorption. Therefore, the desorption process of FTWR could be described as: The FTW on the FTC surface was first desorbed into the solvent due to the principle of compatibility; then, since the interaction between FTW molecules is greater than the force of the pores in the FTC on the FTW molecules. It also gradually desorbs into the solvent to obtain RFTW and WFTC. Hydrogen bonds and other strong interaction forces are the key to the desorption process.

This research solved the problem of high economic cost and waste of resources for FTWR as a hazardous waste disposal and provided an economical and feasible method for the resource utilization of FTWR. The high-quality wax recovered in the process can be further purified and used in other industrial fields to achieve the goal of waste recycling. In addition, this study provided theoretical guidance for a better understanding of the wax desorption process in FTWR.

**Author Contributions:** Investigation, Y.Z. and B.X.; funding acquisition, Y.X.; supervision, Z.L.; writing—original draft, L.L. All authors have read and agreed to the published version of the manuscript.

**Funding:** This research was supported by the National Key Research and Development Project of China during the 13th Five-Year Plan Period (2017YFB0602505).

**Conflicts of Interest:** The authors declare no conflict of interest.

## Nomenclature

| | |
|---|---|
| BET | Brunauer–Emmett–Teller |
| BJH | Barrett–Joyner–Halenda |
| CCD | Central Composite design |
| FT-IR | Fourier transform infrared spectroscopy |
| FTS | Fischer-Tropsch synthesis |
| FTC | Fischer-Tropsch catalyst |
| FTW | Fischer-Tropsch wax |
| FTWR | Fischer-Tropsch wax residue |
| GC-MS | Gas chromatography-mass spectrometer |
| RFTW | recycle Fischer-Tropsch wax |
| RSM | Response Surface Methodology |
| SEM | Scanning electron microscopy |
| WFTC | waste Fischer-Tropsch catalyst |
| XRD | X-ray diffraction |

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
