# Peer review of "Wax Separated Effectively from Fischer-Tropsch Wax Residue by Solvent Desorption: Thermodynamic and Kinetic Analysis"

_applsci, doi:10.3390/app11167745_

Round 1

Reviewer 1 Report

There are a few notes on writing the article.
1. There are many abbreviations in the article. For a better understanding of the results presented, it would be better if the authors sometimes wrote a transcript of these abbreviations in the text. Besides, some abbreviations need to be deciphered, namely RSM and CCD-RSM (lines 13, 74), WFTC (line 409).

2. The phrase “Shimadzu Corporation (Japan) used…” sounds rather strange.

3. In the Fig.1c, what does it mean “RFTE”?

4. Is a high accuracy of the liquid-solid ratio (line 240) important for practical applications?

5. It is not necessary to mention Fig 5 (line 281) before Fig. 4.

6. Figures 6 and 10 captions are too laconic.

7. What hydrogen bonds or chemical bonds can exist in the desorption system?

8. What is the difference between Fig. 7 c and d? This should be reflected in the figure caption.

9. Why there is a capital D in an expression “…particle Diffusion…” (line 380)?

10. What is a crystal plane (20) ? (line 420).

11. It is rather strange that such a complex system as RFTW has a crystal structure of (CH2)x (line 421). Taking into account the complex composition of the system, one could assume that this is a mixture of crystalline and amorphous phases.

12. What does the word "respectively" refer to? (line 430).

Author Response

Dear Reviewer,

Thanks for your constructive suggestions which help us improve the quality of the paper both in depth and English. According to these comments, a careful check and modification on this manuscript have been made. Efforts are also made to correct the mistakes and improve the quality of the manuscript. The revision is made with a point-to-point response to your comments as follows. Please see the attachment.

Sincerely yours,

Ms. Ling Li

Reviewer 2 Report

The authors presented comprehensive and original research on the use of petroleum ether to separate and recover pure stocks from Fischer-Tropsch wax residues. They focused on finding the best conditions for the extraction process. They obtained a very high process efficiency, exceeding 80%. The desorption mechanism proposed by the authors does not raise any doubts. It has been verified with many analyzes such as BET, BJH, SEM and others. I agree that the developed process should find industrial application.

The title of the work is satisfactory. The research topic was precisely defined. The problem was clearly presented. The introduction and the rest of the article are in the right proportions. The results and conclusions are clear and concise.

The literature review was carried out correctly, with the majority of references from the last five years.

The year is missing in the reference 31.

Author Response

Dear reviewer,

Thank you for your constructive suggestions to help us improve the depth of the paper and the quality of English. Based on these opinions, the manuscript was carefully checked and revised. Also strive to correct errors and improve the quality of manuscripts. The revision is made through point-to-point responses to your comments, as shown below. Please see attachment.

Sincerely,

Ms. Li Ling

Reviewer 3 Report

The manuscript "Wax separated effectively from Fischer-Tropsch wax residue by solvent desorption: thermodynamic and kinetic analysis" was well written and presented several interesting results regarding the use of Wax residue from FT process. The results were presented in a clear way. However, I would like to suggest minor modifications: (i) The authors should define the meaning of RSM ; (ii) line 74 - the meaning of CCD is missing (iii) Table 2 - shouldn't the unit of Liquid-solid ratio be L/g (or L/kg)? (iv) line 174 - the authors should replace the word actual for experimental (v) line 180 - the term studentized sounds akward. Is it possible to replace for another word? (vi) in several points of the text the units appear together with the number, e.g. 29nm, 1h.

Author Response

Dear Reviewer,

Thanks for your constructive suggestions which help us improve the quality of the paper both in depth and English. According to these comments, a careful check and modification on this manuscript have been made. Efforts are also made to correct the mistakes and improve the quality of the manuscript. The revision is made with a point-to-point response to your comments as follows.

Sincerely yours,

Ms. Ling Li
